# Children with an Anorectal Malformation Going to Primary School: The Parent’s Perspective

**DOI:** 10.3390/children10060924

**Published:** 2023-05-24

**Authors:** Cunera M. C. de Beaufort, Joep P. M. Derikx, Marijke E. Voskeuil, Josef Atay, Caroline F. Kuijper, Sjoerd A. de Beer, Justin R. de Jong, Arnout de Bos, Svenja Vennink, L. W. Ernest van Heurn, Ramon R. Gorter

**Affiliations:** 1Department of Pediatric Surgery, Emma Children’s Hospital Amsterdam UMC, Location University of Amsterdam, Meibergdreef 9, 1105 AZ Amsterdam, The Netherlands; c.m.debeaufort@amsterdamumc.nl (C.M.C.d.B.); m.e.voskeuil@amsterdamumc.nl (M.E.V.); j.atay@student.acta.nl (J.A.); s.a.debeer@amsterdamumc.nl (S.A.d.B.); j.r.dejong@amsterdamumc.nl (J.R.d.J.); e.vanheurn@amsterdamumc.nl (L.W.E.v.H.); rr.gorter@amsterdamumc.nl (R.R.G.); 2Amsterdam Gastroenterology and Metabolism Research Institute, 1105 AZ Amsterdam, The Netherlands; 3Amsterdam Reproduction and Development Research Institute, 1105 AZ Amsterdam, The Netherlands; 4Department of Pediatric Urology, Emma Children’s Hospital Amsterdam UMC, Location University of Amsterdam, Meibergdreef 9, 1105 AZ Amsterdam, The Netherlands; c.f.kuijper@amsterdamumc.nl; 5Vereniging Anusatresie, 1273 ST Huizen, The Netherlands; arnout@anusatresie.nl (A.d.B.); voorzitter@anusatresie.nl (S.V.)

**Keywords:** anorectal malformations, fecal continence, primary school, parental experiences

## Abstract

Background: Continence problems occur often in children with anorectal malformations (ARM). The aim of this study was to evaluate parental experiences with toilet facilities at Dutch primary schools and their experience with how schools deal with ARM children. Methods: This survey was developed in collaboration with the national patient advocacy group (PAG). Recruitment for participation was conducted by the PAG (email listing and social media) and one expertise center for ARM. Participants were parents of school-attending ARM children aged 3 to 12 years. Results: Sixty-one participants (31.9%) responded to the survey. The median age of the children was 7.0 years (IQR 5.0–9.0). Schools were often located in a village (63.9%) and encompassed 100–500 children (77.0%). In total, 14 parents (23.0%) experienced difficulties in finding a primary school. Experiences with the school were described as solely positive (37.7%), solely negative (9.8%), positive and negative (34.4%), and neither positive nor negative (16.4%). Regarding school toilet facilities, 65.6% of the toilets were reported clean and 78.7% were easily accessible. Conclusions: About 25% of parents reported difficulties in enrolling their children into primary school, and 45% reported negative experiences. This highlights the need for improved guidance and the optimization of education in schools when dealing with ARM children.

## 1. Introduction

Anorectal malformations (ARM) are a rare group of congenital colorectal disorders, affecting approximately 1 per 5000 live births yearly [1,2]. Different types of ARM exist, and they can be classified according to the Krickenbeck classification [3]. Almost all types of ARM require surgical reconstruction in early childhood [1]. However, despite this surgical reconstruction, urinary and defecation problems (i.e., soiling, incontinence, and constipation) can occur in a large proportion of patients with ARM (25% to 60%, depending on the type of ARM) [4,5,6]. These defecation problems might negatively influence their quality of life, as well as result in difficulties for their parents when choosing a primary school for their children [7,8]. Patients with ARM often reached fecal continence at a later age compared to children without ARM [9]. Accurate guidance toward reaching fecal continence, potty training, and going to the bathroom in a timely manner is of great importance when optimizing long-term bowel- and bladder outcomes for patients with ARM. Additionally, procrastination is not desired due to the risk of developing urinary tract infections (in case of voiding postponement) and megarectum and/or megasigmoid (in case of fecal suspension behavior) [10,11]. 

Almost a quarter of the children with ARM that attend primary school might need help and/or assistance when going to the toilet or with changing their clothes in case of accidents or defecation problems during school hours [9,10]. However, most teachers and/or schools are not prepared enough to make sure these children receive the care they need. Consequently, children with ARM might be in the classroom after fecal incontinence accidents with other children surrounding them. Bullying behavior by other children might be provoked due to the foul odor of incontinence, resulting in potentially negative consequences on psychosocial aspects and decreased quality of life in children with ARM [12,13,14]. A study by Judd-Glossy et al. reported that adult patients with a past medical history of ARM commented on the lack of guidance from school during their attendant years [15]. Furthermore, in our opinion, in clinical practice, the parents of patients with ARM often report problems relating to school toilet facilities. The problems mentioned include demands regarding fecal continent, poor bathroom facilities and poor guidance from schools (teachers) regarding potential problems related to ARM in these children. Therefore, the aim of this questionnaire study was twofold. First, we aimed to evaluate parental experiences with toilet facilities at primary schools in the Netherlands for children with ARM aged 3 to 12 years. Second, we evaluated parental experiences on how primary schools deal with children with ARM. 

## 2. Materials and Methods

### 2.1. Study Design

This study was of a cross-sectional design (i.e., a questionnaire). The survey was opened on 21 November 2022, and closed on 1 March 2023. The Schoolproject questionnaire was developed through multiple consensus meetings by a team of medical specialists (pediatric surgeons (*n* = 5), pediatric urologists (*n* = 1), nurse specialists (*n* = 1)) and patient representatives from the national patient advocacy group (PAG) for ARM (‘Vereniging Anusatresie’, *n* = 2). No formal qualitative analysis was performed, but data on parental experiences and final comments were assessed independently by members of the team of experts from both the Amsterdam UMC (CdB/RG), as well as the patient organization (AdB/SV). After an initial assessment, a meeting was held to discuss the results. In case of conflict, a third researcher (CK) was consulted for the final decision. The language in the questionnaire was Dutch. For publication purposes, the questionnaire was translated into English with the help of a native English-speaking researcher (CdB).

### 2.2. Questionnaire

The questionnaire comprised of two main domains. In the first domain, general information (e.g., age of the child, number of schools visited before school choice, school description (i.e., location and size), difficulties in enrolling the child into school, and requirements regarding the child’s potty training) were collected. The second domain comprised of questions on the parent’s experience with the toilet facilities at primary schools for their children (e.g., positive and/or negative experiences, toilet facilities, number of toilets for their children, and specific facilities with respect to toileting). To ensure the anonymity of the data entered into the questionnaire, no questions regarding sex or type of ARM were included. The complete questionnaire can be found in Appendix A.

### 2.3. Participants

The intended participants were parents of school-attending patients with ARM, 3 to 12 years of age. The recruitment of potential participants was conducted by Amsterdam UMC (*n* = 98), an expertise center for ARM, and Vereniging Anusatresie (*n* = 93) separately using a mail listing and a social media account. Due to privacy regulations, patient listings from Amsterdam UMC and Vereniging Anusatresie were not openly available. Therefore, parents could potentially be approached twice. In case this happened, parents were asked to only fill out the survey once. Two months after the first invitation, a reminder for participation was sent by both Amsterdam UMC and Vereniging Anusatresie. The survey was distributed on paper and could also be accessed online through Google Forms. 

### 2.4. Statistical Analysis

Statistical analysis was performed using IBM SPSS Statistics for Windows, Version 28 (IBM Corp., Armonk, NY, USA). Descriptive statistics were used for the analysis of baseline characteristics and survey questions. These were reported as proportions and percentages for binary or categorical variables and as the mean with standard deviation (SD) or a median with an interquartile range (IQR) for continuous variables as appropriate. To test differences in binary or categorical data, Chi-square was used. Toilet facilities (i.e., cleanliness, accessibility, and the presence of toilet assistance) were assessed with univariable analysis as potential predictors for negative experiences toward school toilet facilities. In addition, toilet facilities (i.e., cleanliness, accessibility, the presence of toilet assistance, and trouble enrolling child into school) were assessed with univariable analysis as potential predictors for positive experience towards school toilet facilities. Variables were subsequently selected for multivariable logistic regression analysis and were reported as the odds ratio (OR) with a corresponding 95% confidence interval (CI). A *p*-value below 0.05 was considered statistically significant. Missing or unknown data were described.

## 3. Results

### 3.1. General Characteristics

In total, 61 parents entered the School Project questionnaire (Figure 1). In the eight questionnaires that were completed by less than 90%, answers were most often missing to questions on the aspect of toilets for patients with disabilities. This was because, if available for their children, parents indicated that they were not aware of the qualities of these toilets. The median age of the children whose parents had entered the questionnaire was 7.0 years (IQR 5.0–9.0). Schools were most often located in a village (*n* = 39, 63.9%) and comprised 100 to 500 children (*n* = 47, 77.0%). No statistical differences in general characteristics were identified between the schools located in villages and cities. An overview of general characteristics can be found in Table 1.

### 3.2. Questions Regarding School Choice and Toilet Facilities

In total, 33 parents (54.1%) visited one school before their definitive choice, 15 parents (24.6%) visited two schools before their choice, and 13 parents (21.3%) visited three schools or more before their choice of primary school. Some 38 parents (62.3%) reported schools that had requirements regarding the child’s fecal continence, and 14 parents (23.0%) had difficulties when entering their child into primary school.

The median number of toilets available in schools for children was 2.0 (IQR 2.0–3.0) per classroom. These toilets were reported as clean in 40 schools (65.6%) and easily accessible in 48 schools (78.7%). Overall, the opinions on toilet quality varied widely, as all kinds of examples were mentioned ranging from good or sufficient quality to harrowing with a foul odor, dirt, and a lack of privacy. In addition, toilets for patients with disabilities were present in 30 schools (49.2%). In schools with a toilet for patients with disabilities, 22 schools (73.3%) allowed children with ARM to use them. These toilets for patients with disabilities were reported as clean in 23 schools (76.7%) and easily accessible in 26 schools (86.7%). 

### 3.3. Parental Experiences with Schools for Children with ARM

In total, 53 children (86.9%) used the toilet in school. Reasons for not using the toilet in school varied (e.g., uncomfortable, dislike of the school toilet, and the use of urinary catheterization and colostomies). In 48 schools (78.7%), arrangements were made regarding attending the toilet for children with ARM, of which in 13 schools (27.1%), the arrangements were already in place, and in 8 schools (16.6%), the arrangements were specially made for the child with ARM. Moreover, 21 children (34.4%) obtained assistance and/or help with going to the toilet and/or stoma care from different caregivers (e.g., parents, teachers, nurses, and healthcare facilities).

Parental experiences with school were described as solely positive (*n* = 23, 37.7%), solely negative (*n* = 6, 9.8%), both positive and negative (*n* = 21, 34.4%), or neither positive nor negative (*n* = 10, 16.4%). Positive experiences were reported by 44 parents (72.1%), whereas 27 parents (44.3%) reported negative experiences toward the schools and their toilet facilities that their children with ARM attended. An overview of the reasons for positive and negative parental experiences can be found in Table 2. In multivariable analysis, a clean toilet was identified as a protective factor for negative experiences (i.e., clean toilets were associated with a lower chance of negative experiences, OR 0.18, 95% CI 0.05–0.58, *p* = 0.005, Table 3). In addition, the presence of toilets for patients with disabilities led to higher odds of positive experiences (OR 3.86, 95% CI 1.10–13.54, *p* = 0.035, Table 4). 

### 3.4. Parental Comments

Twenty-nine parents (47.5%) indicated additional comments in response to the survey questions. The most frequently mentioned issues were the parents’ desire that needs concerning going to the toilet in school could be arranged more easily for the future and that there could be more information in schools about ARM. For example, the better assistance of school personnel during toilet visits was deemed very useful instead of parents having to come to school to clean the child their selves (resulting in stress for the parents due to the combination of work and childcare). In addition, parents asked for guidance from the treating hospital that could be provided to teachers and schools so that they could be better informed about the condition of children with ARM. Moreover, parents made suggestions to offer additional information to the school from the treatment hospital and addressed the importance of going to the toilet in a timely manner. Table 5 provides a main theme summary of parental requests.

## 4. Discussion

This study provides an overview of parental experiences regarding school toilet facilities and potential problems when looking for a primary school for their child when suffering from ARM aged 3 to 12 years. About a quarter of the parents reported difficulties in enrolling in a primary school, and almost half reported negative experiences. In multivariable analysis, a clean toilet was found to be associated with a lower chance of negative experiences, whereas the presence of a toilet for patients with disabilities in schools was significantly associated with positive experiences. Almost half of the parents had additional requests, amongst others, regarding the education of teachers and schools on the condition of ARM. 

In multivariable logistic regression, clean toilets were identified to be associated with less negative parental experiences. However, it would also be of great interest to investigate the influence of clean toilets on the experiences of children with ARM. In a previously published study on almost 20,000 Danish school-going children, the majority of children (without any colorectal or urological problem) were found to have dissatisfaction towards school toilets, resulting in an association with bladder and bowel dysfunctions in these children [16]. In addition, the study performed by Vernon et al. showed that the majority of included children reported that aspects of school toilets were harrowing, and at least one-third of the children avoided using the toilet when they needed to defecate [17]. However, these studies were performed among healthy children without chronic underlying conditions such as ARM. Moreover, children with ARM are at greater risk of developing bladder and bowel problems compared to the regular pediatric population [18]. Therefore, clean toilets might be of even greater importance in preventing the development of these problems in children with ARM [13,15].

In this study, different experiences were reported by parents who filled out the survey. In total, 27 parents (44.3%) reported negative experiences, of whom 6 reported negative experiences only. The remaining 21 parents also reported positive experiences with toilet facilities in primary schools for their children with ARM. Unfortunately, despite the importance of this topic, examples in the literature are scarcely available regarding parental experiences in primary schools and their toilet facilities, and only a few studies were published on the experiences of children and primary schools [19]. For example, the studies published by Lum et al. described the school experiences of children and adolescents with a chronic disease, showing mixed experiences and outcomes that worsened compared to children without a chronic disease, though parental experiences were not described [15,20,21]. In addition, studies on patients with (corrected) congenital heart disease were also published to assess the impact of having a chronic disease on school and academic performances, showing the added value of psychological support and guidance throughout their school career [22,23]. This might also be applicable to patients with ARM, as in this study, as some parents reported bullying by other children because of the condition, potentially resulting in decreased quality of life.

Over half of the parents (62.3%) reported that schools had demands regarding the fecal continence of the children that were registered. However, in the Netherlands, schools are not allowed to refuse children that are not yet fully continent. This is especially important considering the fact that parents start searching for schools when their children reach the age of 3 years. Moreover, children are allowed to enter primary school (so-called kindergarten, groups 1 and 2) starting from the age of 4 years. When a child reaches the age of 5 years, official compulsory education starts. Furthermore, toilet training is an important aspect of the general development of children, with and without ARM. Within the International Civic and Citizenship Educations Study (ICCS), the age at which a child was considered ‘incontinent’ was from the age of 5 and older [24]. However, no consensus has been reached on the optimal timing of potty training [25]. For children with ARM, adequate guidance in potty training and going to the bathroom in a timely matter is of the utmost importance to prevent the development of bladder problems and potential megarectum and/or –sigmoid; however, these are also influenced by the quality of life, psychosocial aspects, and educational level [11,15,26,27]. Future research should be performed to investigate patient empowerment and the needs of children with ARM and their parents to optimize potty training facilities in primary schools and prevent parents from experiencing difficulties when enrolling their child with ARM into school [28].

To our knowledge, this is the first study to assess the experience of parents on toilet facilities in primary schools for children with ARM. However, the findings of this study should be interpreted in light of some limitations. First, both Amsterdam UMC and the patient advocacy group “Vereniging Anusatresie” approached patients for participation. However, because of privacy law reasons, it was not possible to find out whether any patients were approached twice, and therefore, the response rate of the survey (32%) might be underestimated. For future studies, this is an important issue to solve. Second, due to the subject of the survey, it could be that the vast majority of responses were from parents with bad experiences and therefore had the urge to share them. Future studies should, therefore, be performed in larger cohorts to overcome this potential negative selection bias and to evaluate the extent of these problems with toilets in primary schools for children with ARM. Additionally, this survey should also be distributed amongst the parents of children without ARM (i.e., ‘healthy’ control group as well as children with other colorectal disorders such as Hirschsprung’s disease or inflammatory bowel disease). Furthermore, further research is also needed to assess the specific needs of parents with children diagnosed with ARM toward parental guidance and education for teachers and primary schools. In order to do so, within our study group, we plan to perform a study that aims to develop instruments (e.g., a website or an application) that enables patients, parents, teachers and schools to gain knowledge about ARM and school-related problems. Finally, in future studies, the description of the parents and families should be investigated, since it could be informative what the status of the parents is, are they single or married, what is their age, do they have a job? This information could also be interesting, in addition to the gathering of information about the whole family, how are they functioning (i.e., including interaction with other siblings), whether there is a safety net, what is the social status of the family, and what does the financial situation look like? Moreover, to identify which schools ‘do better’, more information should be obtained regarding the schools (e.g., private or governmental funded), and the experiences of teachers should also be investigated. To assess the entire situation around these children with ARM, it might be interesting to further investigate all the abovementioned aspects.

## 5. Conclusions

In conclusion, experiences from the parents of children with ARM regarding school toilet facilities varied between positive and negative, and there was a need for educating schools about the condition of ARM. About 25% of the parents reported difficulties when enrolling their child in a primary school, and 45% reported negative experiences. This highlights the need to improve guidance during the school attendance period and to optimize the education of schools when dealing with children with ARM.

## Figures and Tables

**Figure 1 children-10-00924-f001:**
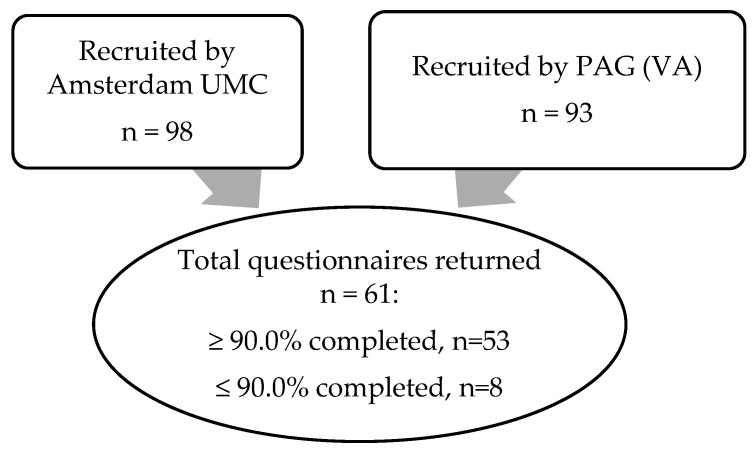
Flowchart of participants (recruited and final entered into the questionnaire).

**Table 1 children-10-00924-t001:** General characteristics.

	Village Schools *n* (%)	City Schools *n* (%)	*p*-Value
Survey completed ≥90%	33 (84.6)	20 (90.9)	0.484
Size			
<100 children	7 (17.9)	4 (18.2)	0.982
100 to 500 children	30 (76.9)	17 (77.3)	0.975
>500 children	2 (5.2)	1 (4.5)	0.919
Admission requirements	24 (61.5)	14 (63.6)	0.970
Experiences			
Positive	29 (74.4)	15 (68.2)	0.807
Negative	18 (46.2)	9 (40.9)	0.807
Both positive and negative	14 (35.9)	7 (31.8)	0.843
Neither positive nor negative	6 (15.4)	4 (18.2)	0.717
Total	39 (63.9)	22 (36.1)	
	median (IQR)	median (IQR)	
Age of child, years	7.0 (4.5–9.0)	7.5 (5.0–9.0)	0.648

*n* = number, IQR = inter quartile range.

**Table 2 children-10-00924-t002:** Overview on reasons for positive and negative parental experiences.

**Positive**	Various ways of help from teachers in case of an accident
	The possibility of using the teachers’ toilet
	Proper arrangements that the child could go to the toilet at all times
	Help from teachers in case of an accident
	The child puts something on the desk when they have to go to the bathroom, so that children cannot use the toilet at the same time.
	A clear explanation of the condition ARM to the other children in the classroom
**Negative**	Delay of going to the toilet (procrastination of urine and/or feces)
	Not being allowed to go to the toilet
	Accidents requiring parents to come into school to change underwear or clothes of their child
	Bullying by other children about incontinence or the condition (ARM)

**Table 3 children-10-00924-t003:** Uni- and multivariable logistic regression for the association between toilet facilities and negative experiences.

	Univariable		Multivariable *	
	OR (95% CI)	*p*-Value	OR (95% CI)	*p*-Value
Toilet status				
Not clean	Ref		Ref	
Clean	0.17 (0.05–0.58)	**0.005**	0.17 (0.05–0.58)	**0.005**
Accessibility				
Not easy	Ref		Ref	
Easy	0.51 (0.14–1.84)	0.304	0.42 (0.11–1.62)	0.209
Toilet assistance				
Not present	Ref		Ref	
Present	0.48 (0.16–1.43)	0.475	0.54 (0.16–1.81)	0.535

Bold in univariable and multivariable analysis indicates statistical significance (*p* < 0.05). * Adjusted for school location (i.e., village, and city) and size (i.e., less than 100 children, between 100 and 500 children, and more than 500 children).

**Table 4 children-10-00924-t004:** Uni- and multivariable logistic regression for the association between toilet facilities and positive experiences.

	Univariable		Multivariable *	
	OR (95% CI)	*p*-Value	OR (95% CI)	*p*-Value
Toilet status				
Not clean	Ref		Ref	
Clean	2.01 (0.61–6.61)	0.251	1.99 (0.60–6.58)	0.258
Accessibility				
Not easy	Ref		Ref	
Easy	2.40 (0.64–9.09)	0.197	2.45 (0.63–9.44)	0.194
Disability toilet				
Not present	Ref		Ref	
Present	3.67 (1.07–12.62)	**0.039**	3.86 (1.10–13.54)	**0.035**
Trouble enrolling				
Not present	Ref		Ref	
Present	0.37 (0.10–1.32)	0.125	0.36 (0.10–1.34)	0.362

Bold in univariable and multivariable analysis indicates statistical significance (*p* < 0.05). * Adjusted for school location (i.e., village, and city) and size (i.e., less than 100 children, between 100 and 500 children, and more than 500 children).

**Table 5 children-10-00924-t005:** Main theme summary of parental requests.

**Personal**
Information/Education
Lack on information in schools regarding ARM
Assistance
Personal assistance was offered by school in case of accidents (parental and/or nurse access)
Medical characteristics
Individual achieved continence was of influence on parental experiences
**School**
Policy
School policy was strict toward children who were not potty trained
Opportunities
Arrangements could be made between parents and school
Facility
School offered specific facilities for child with ARM (e.g., access to toilet for patients with disabilities)

## Data Availability

Requests for data sharing will be considered by the study steering group upon written request to the corresponding author.

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
