# Peer review of "Children with an Anorectal Malformation Going to Primary School: The Parent’s Perspective"

_children, 2023, doi:10.3390/children10060924_

Round 1
Reviewer 1 Report
For school children with bowel dysfunction and fecal incontinence proper and good toilet facilities with easy an accessibility is very important for the child´s well-being and quality of life.
The present study is a questionnaire about parents’ satisfaction with their child’s toilet facilities at school in children operated for ARM and in the age of 3-12 years. Do children in Holland really start in school at the age of 3 years? This needs an explanation.
The patients were recruited from two pediatric surgical centers with a total of 201 eligible patients and only 61 entered the study. An important reason for the low response rate might be all the GPDR-regulations. This calls for a comment – maybe an editorial comment. All these regulations hinder popper health care research which may result in a burden not only for the patients but also for the society!
The questionaries were constructed in co-cooperation with patient’s representative which is a strength.
The results showed that easily accessible and clean toilets was the most important factor for parents to choose a specific school and was associated with high degree of parent satisfaction. Also, assistance from an adult was important This is not surprisingly to my opinion but important to state, anyway. It would have been interesting to include a control group to se if this would have been different for parents to healthy children? How can one secure proper and clean toile facilities in a world of economic restrictions allocated to public schools observed in many western countries. Were any of the schools in the present study private or funded by any means or were they all purely governmental. Why were some schools better than others.
Another important finding was, that the parents requested more information from healthcare professionals to schools and schoolteacher to improve the understanding of the disease and the needs for toilet facilities and assistance from adults during school-time. Do the authors have any specific plan or suggestion to manage this in the future. Must it be driven by patient organizations or health care professionals?
Reviewer 2 Report
The authors stated that potential problems related to anorectal malformations(ARM) in school age children are demands due to fecal incontinence, such as poor bathroom facilities and poor guidance from school (teachers), as the backgrounds. Therefore, they aimed in this study to evaluate the parental experiences with toilet facilities at primary schools for children with ARM, aged 3 to 12 years, in the Netherlands, and to evaluate the parental experiences on how primary schools were dealing with such children. My concerns are as follows.
1. I wonder whether the children attending primary schools are usually aged 3 to 12 years in the Netherlands, although the primary school age might begin at five or six years in many developed countries.
2. Children with the low-level ARM rarely have issues related fecal incontinence. It would be more appropriate to investigate ARM children separately based on the type of ARM.
3. The actual questionnaire is shown in Supplementary Material S1. I wonder whether each question would be better to be included as a list in the main text or in a table, not as a supplement.
4. Only overview on reasons for positive and negative parental experiences is shown in Table 2, and the cleaness, accessibility, and assistance of toilet facilities are shown in Tables 3 and 4. I wonder whether more details of parental requests shown in Supplementary Material S2 would be better to be summarized and included as important data of this survey in the main text.
5. In order to clarify appropriate toilet facilities and improve guidance for ARM children at primary schools, it would be important to survey the same questionnaire from school teachers' side. As noted in Discussion “it could be that the vast majority of responses are from parents with bad experiences”, the data shown in Supplementary Material S2 might be just subjective complaints of the guardians, not objective and concrete data reflecting true conditions of the children at primary schools.
6. I wonder whether data shown in this manuscript could be useful and informative for those caring ARM children abroad, not in the Netherlands.
Nil
Reviewer 3 Report
The introduction is sufficient, but I suggest that in the introduction you target the data for the Netherlands to which the study refers (e.g. prevalence of children with ARM, types of ARM, etc.)
Line 56, 201, 216, etc. The year in parentheses is unnecessary. It is sufficient to provide a reference.
Section 2.3 is unnecessary since all relevant data are listed at the end of the manuscript.
It is necessary to clearly state more information about the design and validation of the questionnaire itself.
You state that the questionnaire was open from 21.11.2022 to 1.03.2023, and on the other hand, that a reminder to participate was sent after 6 months. Do the bases (Amsterdam UMC, an expertise center for ARM, and Vereniging Anusatresie) refer to the total population of the Netherlands? If not, what proportion do they make up?
In the results, clearly state the data in relation to the age and gender of the children to whom the completed questionnaires referred.
I suggest that you show the implementation of the questionnaires, from the total sent to the total completed, using a flowchart.
Check all results and percentages in detail in the text and tables. For example, the sum of the percentages must be 100. Also, in Table 3, I assume the value 0.005 should have been in bold.
Who, and how many experts, made up the team for the implementation of the qualitative part of the research? How did you come to the conclusions stated in section 3.4.? Also, you must clearly state whether the questionnaire was translated into Dutch. Who translated the questionnaire?
Reading the discussion, it is a pity that in your research you did not simultaneously distribute the questionnaires to the parents of children without ARM. At the same time, they would have a control group. I suspect the results would be similar.
You have well noted a number of limitations of your study. It is a pity that during the design and implementation of the study, you did not react to the mentioned parameters in order to make the results more credible.
I have no major complaints about English.
Round 2
Reviewer 2 Report
1. First of all, the authors failed to response appropriately to my inquiries and comments.2. The aims of this questionnaire study noted in the end of Introduction were to evaluate the parental experiences with toilet facilities at primary schools in the Netherlands for children with anorectal malformations (ARM) and to evaluate the parental experiences on how primary schools deal with children with ARM. Without showing concrete and objective data regarding toilet facilities at primary schools in the countries, the data shown there were just based on parental subjective consideration and therefore, cannot be analyzed from scientific points of view. Moreover, the data are hard to be considered informative and interesting for medical staff caring the children with ARM in other countries. This manuscript might be better to be presented in the domestic educational journal.
3. The authors showed overview on reasons for positive and negative parental experiences in Table 2. As I described in the last review comments, such simple data presentation can neither be useful nor helpful to improve toilet facilities at primary schools. Because many factors related to children, schools, teachers and guardians would have a significant impact on such parental experiences, more detailed analyses are mandatory.
4. The analyses regarding the association between toilet facilities and negative/positive experiences shown in Tables 3 and 4 have only small meaning due to a lack of clear definition to estimate toilet status and accessibility for the guardians.
Reviewer 3 Report
Although in this form the research has a lot of limitations, the answers to the reviewer's questions and suggestions and their inclusion in the manuscript, the manuscript has been significantly improved. Thank you for the answers and additional clarifications.
The English language requires minor corrections.